# FVD: A NEW METRIC FOR VIDEO GENERATION

**Thomas Unterthiner**[*]
Johannes Kepler University
unterthiner@ml.jku.at

**Sjoerd van Steenkiste**[*]
IDSIA, SUPSI, USI
sjoerd@idsia.ch

**Karol Kurach**
Google Brain
kkurach@google.com

**Raphaël Marinier**
Google Brain
raphaelm@google.com

**Marcin Michalski**
Google Brain
michalski@google.com

**Sylvain Gelly**
Google Brain
sylvaingelly@google.com

## ABSTRACT

Recent advances in deep generative models have lead to remarkable progress in synthesizing high quality images. Following their successful application in image processing and representation learning, an important next step is to consider videos. Learning generative models of video is a much harder task, requiring a model to capture the temporal dynamics of a scene, in addition to the visual presentation of objects. While recent generative models of video have had some success, current progress is hampered by the lack of qualitative metrics that consider visual quality, temporal coherence, and diversity of samples. To this extent we propose Fréchet Video Distance (FVD), a new metric for generative models of video based on FID. We contribute a large-scale human study, which confirms that FVD correlates well with qualitative human judgment of generated videos.

## 1 INTRODUCTION

Recent advances in deep generative models have lead to remarkable success in synthesizing high-quality images (Karras et al., 2018; Brock et al., 2018). A natural next challenge is to consider video generation. This is a much harder task, requiring a model to capture the temporal dynamics of a scene, in addition to the visual presentation of objects. Generative models of video will enable many applications, including missing-frame prediction (Jiang et al., 2018), improved instance segmentation (Haller & Leordeanu, 2017), or complex (relational) reasoning tasks by conducting inference (Lerer et al., 2016).

While great progress has been made in recent years, video generation models are still in their infancy, and generally unable to synthesize more than a few seconds of video (Babaeizadeh et al., 2017). Learning a good dynamics model remains a major challenge in generating real world videos. However, in order to qualitatively measure progress in synthesizing videos, we require *metrics* that consider visual quality, temporal coherence, and diversity of generated samples.

We contribute *Fréchet Video Distance (FVD)*, a new metric for generative models of video. FVD builds on the principles underlying Fréchet Inception Distance (FID; Heusel et al. (2017)), which has been successfully applied to images. We introduce a feature representation that captures the temporal coherence of the content of a video, in addition to the quality of each frame. Unlike popular

---

[*]Both authors contributed equally to this work while interning at Google Brain.

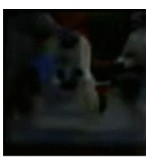 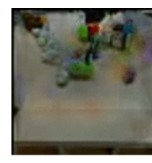 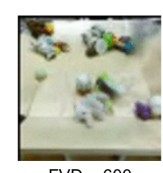 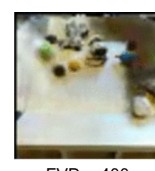 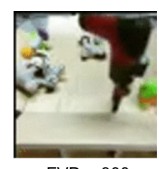 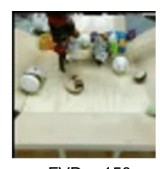

FVD ~2000    FVD ~1000    FVD ~600    FVD ~400    FVD ~300    FVD ~150

Figure 1: Generated videos by various models ranked according to FVD (lower is better).

metrics such as Peak Signal to Noise Ratio (PSNR) or the Structural Similarity (SSIM; Wang et al. (2004)) index, FVD considers a distribution over videos, thereby avoiding the drawbacks of frame-level metrics (Huynh-Thu & Ghanbari, 2012). We contribute extensive experiments to evaluate FVD, including a large-scale human study which confirms that FVD coincides well with qualitative human judgment of generated videos.

## 2 FRÉCHET VIDEO DISTANCE

A generative model of videos must capture the underlying data distribution with which the observed data was generated. The distance between the real world data distribution $P_R$ and the distribution defined by the generative model $P_G$ is a natural evaluation metric. An analytic expression of either distribution is usually unavailable, which prohibits straightforward application of many common distance functions. For example, the popular Fréchet Distance (or 2-Wasserstein distance) between $P_R$ and $P_G$ is defined by: $d(P_R, P_G) = min_{X,Y} \mathrm{E}|X - Y|^2$ where the minimization is over all random variables X and Y with distributions $P_R$ and $P_G$ respectively. This expression is difficult to solve for the general case, but has a closed form solution when when $P_R$ and $P_G$ are multivariate Gaussians (Dowson & Landau, 1982):

$$d(P_R, P_G) = |\mu_R - \mu_G|^2 + \mathrm{Tr}\left(\Sigma_R + \Sigma_G - 2(\Sigma_R \Sigma_G)^{\frac{1}{2}}\right) \tag{1}$$

where $\mu_R$ and $\mu_G$ are the means and $\Sigma_R$ and $\Sigma_G$ are the co-variance matrices of $P_R$ and $P_G$. A multivariate Gaussian is seldom an accurate representation of the underlying data distribution, but when using a suitable feature space, this is a reasonable approximation. For distributions over real world *images*, Heusel et al. (2017) used a learned feature embedding to calculate the distance between $P_R$ and $P_G$ as follows: First samples from $P_R$ and $P_G$ are fed through an Inception network (Szegedy et al., 2016) network that was trained on ImageNet and their feature representations (activations) in one of the hidden layers are recorded. Then the Fréchet Inception Distance (FID; Heusel et al. (2017)) is computed via Eq. 1 using the means and covariances obtained from the recorded responses of the real, and generated samples.

The feature representation learned by the pre-trained neural network greatly affects the quality of the metric. When training on ImageNet, the learned features expose information useful in reasoning about objects in images, whereas other information content may be suppressed. Likewise, different layers of the network encode features at different abstraction levels. In order to obtain a suitable feature representation for videos we require a pre-trained network that considers the temporal coherence of the visual content across a sequence of frames, in addition to its visual presentation at any given point in time. In this work we investigate several variations of a pre-trained Inflated 3D Convnet (I3D; Carreira & Zisserman (2017)), and name the resulting metric the *Fréchet Video Distance (FVD)*[1][2].

The I3D network generalizes the Inception architecture to sequential data, and is trained to perform action-recognition on the Kinetics data set consisting of human-centered YouTube videos Kay et al. (2017). Action recognition requires visual context and temporal evolution to be considered simultaneously, and I3D has been shown to excel at this task. We explore two different feature representations (logits, avg. pool) learned by I3D networks pre-trained on Kinetics-400, and Kinetics-600.

A potential downside in using Eq. 1 is the potentially large error in estimating Gaussian distributions over the learned feature space. Bińkowski et al. (2018) proposed to use the Maximum Mean Discrepancy (MMD; Gretton et al. (2012)) as an alternative in the case of images, and we will explore this variation in the context of videos as well. MMD is a kernel-based approach, which provides a means to calculate the distance between two empirical distributions without assuming a particular form. Bińkowski et al. (2018) proposed to use a polynomial kernel $k(a, b) := \left(a^T b + 1\right)^3$, which we will apply to the learned features of the I3D network to obtain the Kernel Video Distance (KVD).

---

[1]A similar adaptation of FID was used by Wang et al. (2018) to evaluate their *vid2vid* model. Here we introduce FVD as a general metric for videos and an focus on an extensive empirical study.

[2]Code to compute FVD is available at `https://git.io/fpuEH`.

## 3 EXPERIMENTS

In the following we present results of a noisy study, and human study of FVD. Additional experiments that analyze its sensitivity to sample size, and resolution are available in Appendix B & D.

### 3.1 NOISE STUDY

We test how sensitive FVD is to basic distortions by adding noise to the real videos. We consider *static* noise added to individual frames, and *temporal* noise, which distorts the entire sequence of frames. We applied these distortions (details in Appendix A) at up to six different intensities, and computed the FVD and KVD between videos from the BAIR (Ebert et al., 2017), Kinetics-400 (Carreira & Zisserman, 2017) and HMDB51 (Kuehne et al., 2013) data sets and their noisy counterparts. As potential embeddings, we considered the top-most pooling layer, and the logits layer of the I3D model pre-trained on the Kinetics-400 data set, as well as the same layers in a variant of the I3D model pre-trained on the extended Kinetics-600 data set. As a baseline, we compared to a naive extension of FID for videos in which the Inception network (pre-trained on ImageNet) is evaluated for each frame individually, and the resulting embeddings (or their pair-wise differences) are averaged to obtain a single embedding for each video. This "FID" score is then computed according to Eq. 1.

All variants were able to detect the various injected distortions to some degree, with the pre-trained Inception network generally being inferior at detecting temporal distortions as was expected. Figure 2 shows that the logits layer of the I3D model pre-trained on Kinetics-400 has the best average rank correlation with the sequence of noise intensities. An overview of its scores on the noise experiments can be seen in Figure 3.

### 3.2 HUMAN EVALUATION

| Metric | eq. FVD | eq. SSIM | eq. PSNR | spr. FVD | spr. SSIM | spr. PSNR |
|--------|---------|----------|----------|----------|-----------|-----------|
| FVD | N/A | **74.9** % | **81.0** % | **71.9** % | **58.4** % | **63.5** % |
| SSIM | 51.5 % | N/A | 44.6 % | 61.8 % | 51.2 % | 45.9 % |
| PSNR | **56.3** % | 21.4 % | N/A % | 54.1 % | 37.0 % | 44.8 % |

Table 1: Agreement of metrics with human judgment when considering models with a fixed value for a given metric (eq.), or with spread values over a wide range (spr.).

One important criterion for the performance of generative models is the visual fidelity of the samples as judged by human observers (Theis et al., 2016), as a metric for generative models must ultimately correlate well with human judgment. Thus we trained several conditional video generation models, and asked human raters to compare the quality of the generated videos in different scenarios.

**Results** The results of the human evaluation studies can be seen in Table 1, and additional results for KVD and Avg. FID in Appendix C. We find that FVD is the superior choice compared to all other metrics tested. The results obtained for eq. FVD and spr. FVD are of key importance as they determine how users will experience FVD in practice. From the spr. FVD column we can conclude that no other metric can improve upon the ranking induced by FVD, and the eq. FVD column tells us that no other metric can reliably distinguish between good models that are equal in terms of FVD.

On the other hand, FVD is able to distinguish models when other metrics can not (eq. SSIM, eq. PSNR), agreeing well with human judgment (74.9 %, and 81.0 % agreement respectively). Likewise FVD consistently improves on the ranking induced by other metrics (spr. SSIM, spr. PSNR), even though these scenarios are clearly favorable for the metric under consideration.

## 4 CONCLUSION

We introduced the Fréchet Video Distance (FVD), a new evaluation metric for generative models of video, and an important step towards better evaluation of models for video generation. Our experiments confirm that FVD is accurate in evaluating videos that were modified to include static

noise, and temporal noise. More importantly, a large scale human study among generated videos from several recent generative models reveals that FVD consistently outperforms SSIM and PSNR in agreeing with human judgment.

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

## A   NOISE STUDY

To test whether FVD can detect static noise we added one of the following distortions to each frame in a sequence of video frames: (1) a *black rectangle* drawn at a random location in the frame, (2) *Gaussian blur*, which applies a Gaussian smoothing kernel to the frame, (3) *Gaussian noise*, which interpolates between the observed frame and standard Gaussian noise, and (4) *Salt & Pepper noise*, which sets each pixel in the frame to either black or white with a fixed probability. Temporal noise was injected by (1) *locally swapping* a number of randomly chosen frames with its neighbor in the sequence (2) *globally swapping* a number of randomly chosen pairs of frames selected across the whole sequence, (3) *interleaving* the sequence of frames corresponding to multiple different videos to obtain new videos, and by (4) *switching* from one video to another video after a number of frames to obtain new videos. We applied these distortions at up to six different intensities that are unique to each type, e.g. related to the size of the black rectangle, the number of swaps to perform, or the number of videos to interleave.

We conduct the noise study on HMDB (Kuehne et al., 2013), BAIR (Ebert et al., 2017), and Kinetics-400 (Kay et al., 2017). A total of 90% of the available samples (train and test) were used to perform the comparison. A mapping of the different noise intensities to the parameter values of the various noise types that we consider can be seen in Table 2.

| Noise type | Parameter | Int. 1 | Int. 2 | Int. 3 | Int. 4 | Int. 5 | Int. 6 |
|---|---|---|---|---|---|---|---|
| Black rectangle | size relative to image | 15% | 30% | 45 % | 60 % | 75 % | N/A |
| Gaussian blur | sigma of Gaussian kernel | 1 | 2 | 3 | 4 | 5 | N/A |
| Gaussian noise | mixing factor | 15% | 30% | 45 % | 60 % | 75 % | N/A |
| Salt & Pepper | prob. of applying noise | 0.1 | 0.2 | 0.3 | 0.4 | 0.5 | N/A |
| Local swap | nr. of swaps | 4 | 8 | 12 | 16 | 20 | 24 |
| Global swap | nr. of swaps | 4 | 8 | 12 | 16 | 20 | 24 |
| Interleaving | nr. of sequences | 2 | 3 | 4 | 5 | 6 | N/A |
| Switching | nr. of frames until switch | 1 | 2 | 3 | 4 | 5 | N/A |

Table 2: An overview of the different noise intensities used for different noise types.

Figure 2 provides an overview of the correlation of various implementations of FVD (and an FID-based baseline) with the sequence of noise intensities. It can be seen that the logits of the I3D model trained on the Kinetics 400 dataset correlates well with the noise intensities, across a variety of noise types. Its performance can be seen in Figure 3

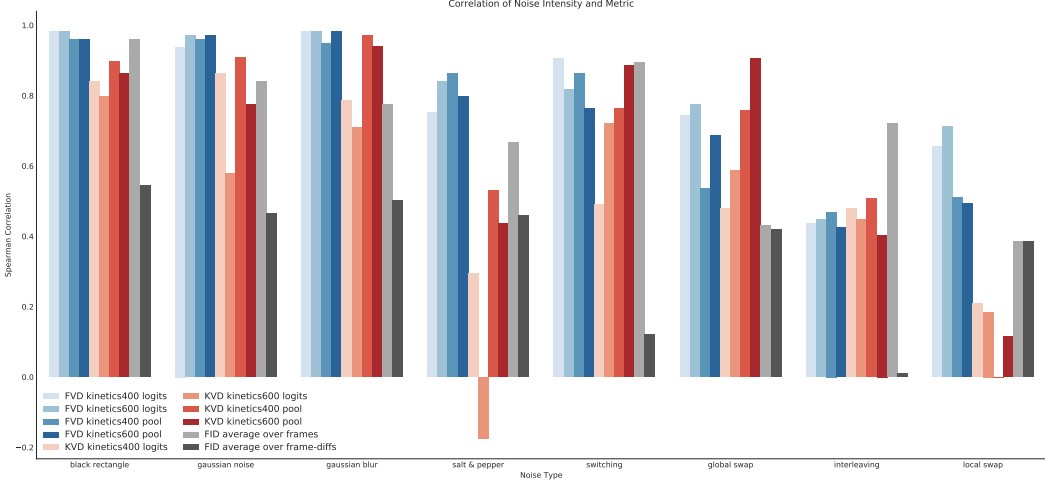

Figure 2: Correlation of the noise intensity and the metric measurements.

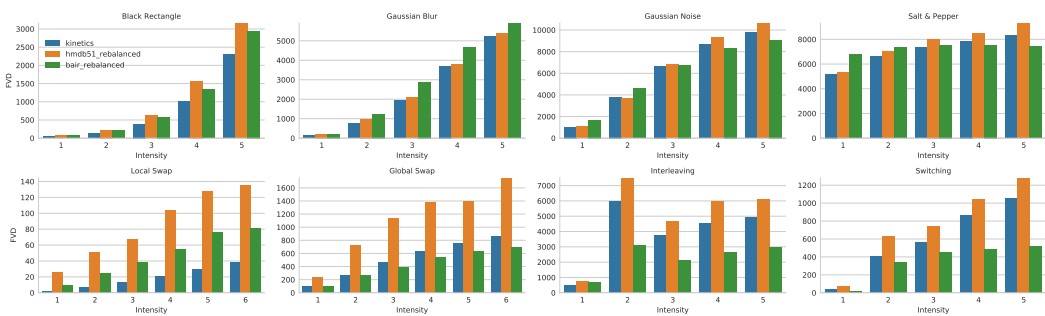

Figure 3: Behaviour of FVD when adding various types of noise to different data sets, using the logits activations of the I3D model trained on Kinetics-400 as embedding.

## B   EFFECT OF SAMPLE SIZE ON FVD

We consider the accuracy with which FVD is able to calculate the true underlying distance between a distribution of generated videos and a target distribution. To calculate the FVD according to Eq. 1 we need to estimate $\mu_R, \mu_G$ and $\Sigma_R, \Sigma_G$ from the available samples. The larger the sample size, the better these estimates will be, and the better FVD will reflect the true underlying distance between the distributions. For an accurate generative model these distributions will typically be fairly close, and the noise from the estimation process will primarily affect our results. This effect has been well-studied for FID (Lucic et al., 2018; Bińkowski et al., 2018), and is depicted for FVD in Figure 4. It can be seen that even when the underlying distributions are identical, FVD will typically be larger than zero because our estimates of the parameters $\mu_R, \mu_G, \Sigma_R$ and $\Sigma_G$ are noisy. It can also be seen that for a fixed number of samples the standard errors (measured over 50 tries) are small, and an accurate comparison can be made. Hence, it is critical that in comparing FVD values across models, one uses the same sample size[3].

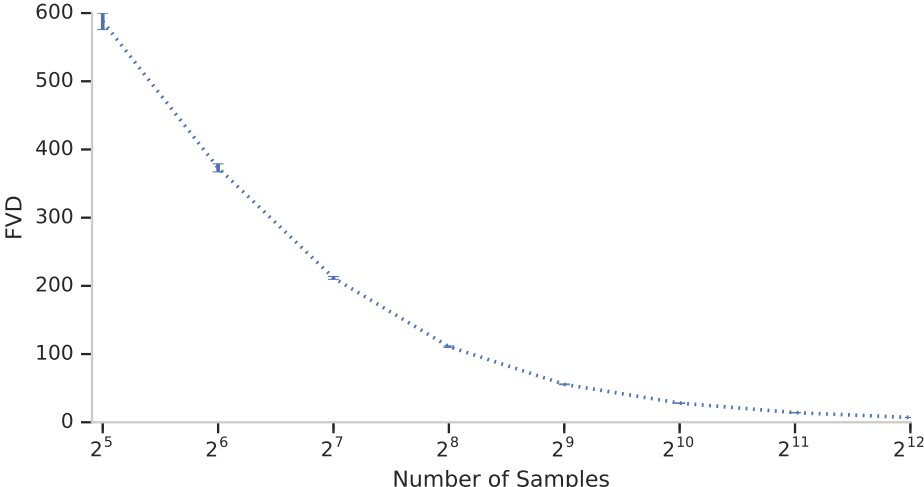

Figure 4: FVD between two non-overlapping subsets of videos that are randomly drawn from the BAIR video pushing data set. Error bars are standard errors over 50 different tries.

---

[3]This has lead to confusion regarding FID in the past, when researchers used different sample sizes in their comparisons (Lucic et al., 2018; Heusel et al., 2017).

## C  HUMAN EVALUATION

We trained CDNA (Finn et al., 2016), SV2P (Babaeizadeh et al., 2017), SVP-FP (Denton & Fergus, 2018) and SAVP (Lee et al., 2018) on the BAIR data set. Using a wide range of possible hyper-parameter settings, and by including model parameters at various stages of training we obtain over 3 000 different models. Generated videos are obtained by combining 2 frames of context with the proceeding 14 output frames. Following prior work we obtain the PSNR and SSIM scores by generating 100 videos for each input context (conditioning frames) and returning the best frame-averaged value among these videos. We consider 256 video sequences (unseen by the model) to estimate the target distribution when computing FVD.

We conduct several human studies based on different subsets of the trained models. In particular, we select models according to two different scenarios:

**One Metric Equal**  We consider models that are indistinguishable according to a single metric, and evaluate to what degree human raters and other competing metrics are able to distinguish these models in terms of the quality of the generated videos. We choose 10 models having roughly equal values for a given metric that are close to the best quartile of the overall distribution of that metric, i.e., the models were chosen such that they are worse than 25 % of the remaining models and better than 75 % of the remaining models as determined by the metric under consideration. We were able to choose models whose values where identical up to the first 4-5 significant digits for each metric.

**One Metric Spread**  In a second setting we consider to what degree models having very different scores for a given metric, coincide with the subjective quality of their generated videos as judged by humans. We choose 10 models which were equidistant between the 10 % and 90 % percentile of the overall distribution of that metric. In this case, there should be clear differences in terms of the quality of the generated videos among the models under consideration (provided that the metric is accurate), suggesting high agreement with human judgment for the metric under consideration in comparison to competing metrics.

For the human evaluation, we used 3 generated videos from each selected model. Human raters would be shown a video from two models, and then asked to identify which of the two looked better, or alternatively report that their quality was indistinguishable. Each pair of compared videos was shown to up to 3 independent raters, where the third rater was only asked if the first two raters disagreed. The raters were given no prior indication about which video was thought to be better. We calculated the correspondence between these human ratings and the ratings determined by the various metrics under consideration.

| Metric | eq. FVD | eq. SSIM | eq. PSNR | eq. KVD | spr. FVD | spr. SSIM | spr. PSNR | spr. KVD |
|---|---|---|---|---|---|---|---|---|
| FVD | N/A | **74.9 %** | **81.0 %** | **63.0 %** | **71.9 %** | 58.4 % | 63.5 % | **63.1 %** |
| SSIM | 51.5 % | N/A | 44.6 % | 43.6 % | 61.8 % | 51.2 % | 45.9 % | 50.2 % |
| PSNR | **56.3 %** | 21.4 % | N/A | 48.8 % | 54.1 % | 37.0 % | 44.8 % | 54.1 % |
| KVD | 40.6 % | 70.4 % | 73.8 % | N/A | 69.4 % | 56.8 % | **63.8 %** | 59.1% |
| Avg. FID | 35.5 % | 71.2 % | 52.0 % | 43.5 % | 62.4 % | **62.7 %** | 57.6 % | 51.2 % |
| raters | 79.3 % | 77.8 % | 84.4 % | 74.3 % | 83.3 % | 69.9 % | 72.5 % | 74.1 % |

Table 3: Agreement of metrics with human judgment when considering models with a fixed value for a given metric (eq.), or with spread values over a wide range (spr.). FVD is superior at judging generated videos based on subjective quality.

Table 3 contains the results of the human evaluation for the KVD and Avg. FID metrics in addition to the results for all other metrics from the main paper. Avg. FID is computed by averaging the Inception embedding for each frame, before computing the Fréchet distance.

- **Avg. FID** We find that Avg. FID performs markedly worse compared to FVD in most scenarios, except on spr. SSIM, where it achieves slightly better performance. It suggests that it is preferential to judge the wide range of videos (sampled from each decile as determined by SSIM) based primarily on frame-level quality. On the other hand, when considering

    videos of similar quality in *eq. SSIM*, we find that judging based on temporal coherence (in addition to frame-level quality) is beneficial and Avg. FID performs worse.

- **KVD** We find that KVD is highly correlated with FVD (spearman: 0.9), although in most scenarios it performs slightly worse than FVD in terms of agreement with human judgment.

In general we may conclude from Table 3 that FVD is the superior choice compared to all other metrics tested. The results obtained for eq. FVD and spr. FVD are of key importance as they determine how users will experience FVD in practice. From the spr. FVD column we can conclude that no other metric can improve upon the ranking induced by FVD, and the eq. FVD column tells us that no other metric can reliably distinguish between good models that are equal in terms of FVD.

Table 3 also reports the agreement among raters. These are computed as the fraction of the comparisons in which the first two raters agreed for a given video pair, averaged across all comparisons to obtain the final percentage. It can be seen that in most cases the raters are confident in comparing generated videos.

# D RESOLUTION OF FVD

While the results in Appendix B demonstrates that FVD results are highly reproducible for a fixed sample size, it does not consider to what degree small differences in FVD can be considered meaningful. To answer this question, human raters were asked to compare videos generated by a randomly chosen model having an FVD of 200 / 400 (base200 / base400) and generated videos by models that were 10, 20, 50, 100, 200, 300, 400, and 500 FVD points worse. In each case we selected 5 models from the models available at these FVD scores and generated 3 videos for each model, resulting in a total of 1 800 comparisons. For a given video comparison, raters were asked to decide which of the two videos looked better, or if they were of similar quality. For each of these pairs, we asked up to 3 human raters for their opinion.

In Figure 5 it can be seen that when the difference in FVD is smaller than 50, the agreement with human raters is close to random (but never worse), and increases rapidly once two models are more than 50 FVD points apart. Hence, it can be concluded that differences of 50 FVD or more typically correspond to differences in the quality of the generated videos that can be perceived by humans.

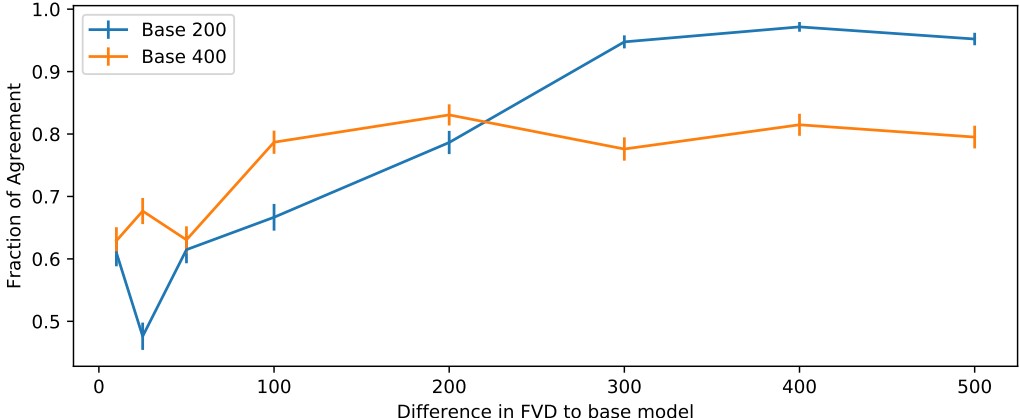

Figure 5: Fraction of human raters that agree with FVD on which of two models is better, as a function of the difference in FVD between the models. Error bars are standard errors, and raters deciding that video pairs are of similar quality are counted as not agreeing with FVD.

