# OpenReview forum: "FVD: A new Metric for Video Generation"
_ICLR.cc/2019/Workshop/DeepGenStruct — DeepGenStruct 2019_

### Official Review · AnonReviewer1 · 2019-04-14
**metric for evaluating video generation**

**Rating:** 4
**Confidence:** 2

**Review:**

The paper extends the FID metric used for evaluating generative sample quality to video generation. They use I3D network that generalises the Inception architecture to sequential data. The paper provides extensive comparison with other baseline method and does a thorough human evaluation to show that their proposed metric has significant consensus with human evaluation.

---

### Official Review · AnonReviewer2 · 2019-04-15
**A new metric to evaluate generative models of video**

**Rating:** 3
**Confidence:** 1

**Review:**

The paper presents a new metric (FVD) for generative models of video. It basically builds on Frechet Inception Distance (FID) which has been proposed for Images and extends it to sequential data such as videos. It captures both the temporal coherence of the content of the video and the quality of each frame. Authors present a thorough evaluation of this proposed metric and show that it correlates with the qualitative human judgements of generated video.

I am not an expert in this area. Still, I felt that the paper could have been presented in a better way. The new metric FVD is not well motivated. The paper is difficult to read, it is written as a summary, and most of experimental setup and evaluated systems are moved to appendices.  Given that the main paper is presented in only 3 pages (with appendices, it is only 8 pages), I  didn't understand the need of appendix sections. First three pages were not inclusive. If accepted, I would recommend to make the paper inclusive of all the details.

---

### Decision · Program_Chairs · 2019-04-19
**Acceptance Decision**

**Decision:**

Accept

**Comment:**

Accepted